# Stress, Resilience, Burnout and Study Hours in Physical Education Pre-Service Teachers—An Explanatory Model about Gender

**DOI:** 10.3390/bs13110946

**Published:** 2023-11-17

**Authors:** Eduardo Melguizo-Ibáñez, Gabriel González-Valero, José Manuel Alonso-Vargas, Rafael Caracuel-Cáliz, Manuel Ortega-Caballero, Pilar Puertas-Molero

**Affiliations:** 1Department of Didactics of Musical, Artistic and Corporal Expression, University of Granada, 18071 Granada, Spain; edumeliba@correo.ugr.es (E.M.-I.); ggvalero@ugr.es (G.G.-V.); pilarpuertas@correo.ugr.es (P.P.-M.); 2International University of La Rioja (UNIR), 26006 Logroño, La Rioja, Spain; 3Faculty of Education Science, Universidad Internacional de Valencia (VIU), 46002 Valencia, Spain; 4Department of Pedagogy, Faculty of Education and Sport Sciences, Melilla Campus, University of Granada, 52005 Granada, Spain; manorca@ugr.es

**Keywords:** mental health, physical education teacher, Spanish educational system, gender

## Abstract

The process of becoming a public teacher in Spain requires a long period of preparation. This long period of preparation has an impact on the psychosocial environment of the candidates. Differences have been observed in the psychosocial area according to gender in pre-service teachers. This research aims to study the relationship between the study hours per day, stress, burnout syndrome and resilience according to gender and to study the differences in the effects according to gender using multigroup equation modeling. A multigroup structural equation analysis has been proposed according to the gender of the participants. Parametric tests were used for the descriptive analysis of the results. The sample consists of 4117 participants, 1363 males and 2754 females. The instruments used to collect the data were a self-made questionnaire, Perceived Stress Questionnaire, Connor-Davidson Resilience Scale and Maslach Burnout Inventory. All the instruments have been validated and adapted to the sample. The data reveal that there are variations in the effects of the variables according to the gender of the participants. In conclusion, it is affirmed that gender is a very important factor in coping with the competitive examination process for state-public-teaching institutions, as well as in avoiding the appearance of disruptive states generated by this preparation process.

## 1. Introduction

The established process of becoming a teacher in the Spanish education system involves passing a series of very demanding tests [1]. The so-called “*Opositores*” are those candidates destined to obtain one of the various posts offered [2]. This rigorous entrance examination consists of a series of tests. The first test is designed to demonstrate the specific knowledge required for the teaching field for which the candidate is applying [3]. It consists of the candidate’s development of a topic from the official syllabus from a number of topics drawn at random by the selection board [3]. The second consists of the preparation, presentation and defence of a teaching unit [3]. The teaching unit must be related to the candidate’s field [3].

Obtaining a permanent Spanish teaching post generates high levels of uncertainty and stress for applicants [4]. This is due to the large number of applicants and the uncertainty generated in the period leading up to the test [4]. The concept of stress is understood as a series of alterations produced in the organism in response to different stimuli [5]. Continuous stress can cause mental fatigue [6]. This fatigue can originate due to the demand for higher-than-normal performance, which leads to the development of mental and physical disorders [6]. The onset of stress occurs in three distinct phases [7]. The first is the alarm reaction [7]. This phase focuses on alerting the body to the effect of a stressor [7]. If this first phase is prolonged over time, it gives way to the resistance phase [7]. This phase focuses on the subject coping with the stressor [7]. The third phase is known as the exhaustion phase [7]. It is characterised by the onset of fatigue and a decrease in motivation towards the task [7].

Performing a task with low levels of motivation leads to a process of emotional and psychological exhaustion known as burnout [8]. It is known as a state characterised by a low commitment to the task and low levels of tolerance [9]. If this state persists over time, physical, psychological and emotional fatigue are observed [9]. The combination of the above three fatigues leads to emotional and psychological exhaustion [9]. Burnout syndrome can also be characterised by depersonalisation, emotional exhaustion and low self-fulfilment [10]. Burnout can have a negative effect on the psychosocial domain [11]. It has been observed that there are key factors in the development of Burnout Syndrome. One of them is gender [12]. García-Martínez et al. [13] and Guerrero-Barona et al. [14] conclude that males tend to have an earlier onset of this disturbing state. This is due to lower levels of emotional training [13,14]. Prolonged exposure to a high level of burnout can be detrimental to physical and mental health [15]. Resilience plays a key role in the prevention of Burnout [16].

Resilience helps to prevent the onset of burnout syndrome [16]. This term is defined as the intrinsic ability of individuals to overcome stressful states [17]. It has been observed that this intrinsic capacity helps to achieve the different academic goals set [17]. It also helps to overcome adverse and stressful situations [17]. In the academic context, resilience has been extensively studied [18,19]. It has been concluded that in the professional setting this intrinsic capacity is used to achieve goals and overcome disruptive emotional states [20]. Similarly, resilience can be conditioned using different variables. Dolev et al. [21] claim that gender plays a key role in resilient behaviour. It has been shown that women show higher levels of resilience than men [21]. In academia, women show higher levels of resilience in coping with negative stimuli [21].

Within the study population, there is little research focused on these three variables. Melguizo-Ibáñez et al. [22] found that levels of resilience, stress and burnout vary according to the speciality of trainee teachers. Research has shown the need to develop resilient behaviours in this population [23]. This is due to the various negative emotional states to which aspiring teachers are subjected [23]. Throughout this preparation process, disturbing attitudes may be experienced that negatively affect a person’s motivation [22]. This can act to the detriment of the preparation process by reducing study hours [22]. Among the factors that buffer stress and burnout syndrome in trainee teachers is the perception of the job as being valuable [23]. Getting a job contributes to eudaimonic well-being [23]. In addition, other variables such as personal resources, teacher identity and support from qualified staff have been identified for the development of resilience [24]. Furthermore, building resilience in trainee physical education teachers has been found to be effective in reducing levels of stress and burnout [24].

The following research questions have been established: Does resilience exert a positive effect on the reduction of stress and burnout syndrome? (O2 and H2) Does resilience act positively on the number of study hours? (O2 and H5) Do stress and burnout syndrome have a negative effect on the number of study hours? (O2, H3 and H4) Are there variations in resilience to burnout syndrome, stress and number of study hours that result as a function of the gender of the applicants? (O1, O2, H1, H6, H7, H8 and H9).

The novelty of this article lies in the need to understand the effect of psychosocial factors on teacher preparation as a function of gender. For this purpose, a multi-group structural equation analysis was carried out according to the sex of the participants. The objectives established are as follows:

**O1.** To study the level of burnout, stress, resilience and study hours as a function of gender.

From this objective, the following research hypotheses are proposed:

**H1.** 
*Aim to observe differences in the levels of burnout, resilience, stress and number of hours of study as a function of the gender of the participants.*


**O2.** To study the differences in the effects according to gender using multigroup equation modeling.

The following research hypotheses are developed from this objective (O2).

**H2.** 
*The impact of resilience on stress will be negative.*


**H3.** 
*Stress will have a negative effect on the hours of study.*


**H4.** 
*Burnout will act negatively on the number of study hours.*


**H5.** 
*Resilience will exert a positive effect on daily study hours.*


**H6.** 
*Female population will show a greater effect of resilience on stress.*


**H7.** 
*The effect of stress on the number of hours of study will be greater for males.*


**H8.** 
*The female pre-service teachers will show a greater effect of burnout on study hours.*


**H9.** 
*The effect of resilience on study hours will be greater for the female population.*


## 2. Materials and Methods

### 2.1. Design and Participants

The study presented here is of a non-experimental, exploratory, descriptive and cross-sectional design. Only one single period of data collection was carried out. The final sample consisted of 4117 applicants. Table 1 presents the distribution of the sample according to the gender and speciality of the applicants. The majority of the sample aspire to get a job in primary education (n = 2318; 56.3%). According to gender, more than two thirds identify themselves as female (n = 2754; 66.9%).

Data collection was applied at a national level. The distribution of the sample according to autonomous community is proposed below: Andalusia (n = 922; 22.4%), Catalonia (n = 158; 3.8%), Community of Madrid (n = 629; 15.3%), Valencian Community (n = 576; 14.0%), Galicia (n = 401; 9.7%), Castille and Leon (n = 254; 6.2%), Basque Country (n = 30; 0.7%), Canary Island (n = 126; 3.1%), Region of Murcia (n = 213; 5.2%), Aragon (n = 73; 1.8%), Balearic Island (n = 31; 0.8%), Extremadura (n = 110; 2.7%), Asturias (n = 113; 2.7%), Navarre (n = 39; 0.9%), Cantabria (n = 69; 1.7%) and La Rioja (n = 14; 0.3%).

With respect to the inclusion criteria, three criteria were established:(1)To have a degree in early childhood and/or primary education,(2)To have another university degree and a master’s degree in teaching.(3)To be an applicant for the different public teaching positions in Spain.

Failure to meet any of these three inclusion criteria resulted in elimination of the participant.

The participants were contacted through different professors from the different Spanish public universities. The data collection instrument was administered virtually. Before starting to fill in the different instruments, the participants signed the informed-consent form. Participants were assured that the data would be treated anonymously and would be used for scientific purposes. Finally, the instruments were answered by the participants themselves.

### 2.2. Instruments and Study Variables

The instruments used have been validated. The versions used are reliable for data collection in this sample. Cronbach’s alpha test was also used to study the degree of internal consistency of the scales and subscales.

An ad hoc questionnaire: The variables of gender (male/female), age and average number of hours of study per day were collected.

Perceived Stress Questionnaire [25]: The Spanish version has been used [26]. It is made of 14 items that are measured on a Likert scale (0 = never, 4 = very often). A value of α = 0.905 was obtained.

Connor-Davidson Resilience Scale [27]: The Spanish version [28] was used. It assesses resilience through 25 items. The dimensions are control and purpose, persistence/tenacity/self-efficacy, adaptability and support networks, control under pressure, and spirituality [28]. Cronbach’s alpha obtained a score of α = 0.916. Cronbach’s alpha for the different subscales is shown below: persistence/tenacity/self-efficacy: α = 0.900; control under pressure: α = 0.897; adaptability: α = 0.945; control and purpose: α = 0.915; and spirituality: α = 0.924.

Maslach Burnout Inventory [29]: The version adapted to Spanish has been used [30]. This inventory consists of 22 items assessed using a three-dimensional perspective: emotional exhaustion (1, 2, 3, 6, 8, 13, 14, 16 and 20), degree of depersonalisation (5, 10, 11, 15 and 22) and personal fulfilment (4, 7, 9, 12, 17, 18, 19 and 21). Cronbach’s alpha evidenced the next values of α = 0.869. Cronbach’s alpha for the different subscales is shown below: emotional exhaustion: α = 0.900; depersonalization: α = 0.865 and personal fulfillment: α = 0.892.

### 2.3. Procedure

This section contextualises the procedure followed during data collection. It also discusses the ethical aspects of the study.

Before starting the fieldwork, a review was carried out. The aim was to find out which instruments were the most reliable for data collection. The PubMed, Scopus and Web of Science databases were consulted to search for studies on this subject. Once the analyzed instruments had been analyzed, we proceeded to search for studies related to psychometric properties. Once the different properties had been analyzed, the final instruments were selected. Once selected, a Google Form document was created. This contained the research objectives and informed consent. The data were collected virtually. This allowed access to a larger number of participants. To ensure that responses were not random, one question was duplicated. For participants who did not match the answer to this question, the person’s participation was eliminated. This resulted in the elimination of 15 participants. A total of 22 participants were eliminated because they did not meet the inclusion criteria. In total, 37 participants were eliminated. This study was conducted during the months of January and June 2022. This coincided with the period leading up to the entrance examination. The research followed the ethical criteria of the Declaration of Helsinki. In addition, participants participated voluntarily and the health and anonymity of the participants was guaranteed.

### 2.4. Data Analysis

IBM SPSS 25.0 was used to perform the comparative analyses. Initially, a study of the sample distribution and homogeneity of variance was carried out. The Kolmogorov–Smirnov test was used. This test revealed a normal distribution. After observing a normal distribution, parametric tests were used. For comparative analysis between two groups, a Levene test was used together with the t-Student. Statistically significant differences were determined using Pearson’s chi-square test (*p* ≤ 0.05). Cohen’s *d* statistic was also used to study the effect size [31]. The effect size is interpreted according to the score obtained: null (≤0.19), low (0.20–0.49), moderate (0.50–0.79) and strong (≥0.80).

IBM SPSS Amos 26.0 was used to perform the exploratory analysis (Objective 2). A multi-group model was run according to the sex of the participants. Each model is composed of nine endogenous and two exogenous variables (Figure 1). For the first type of variables, a causal explanation was put forward based on the degree of the reliability of the measures and the observed associations. For the endogenous variables, error was included (e1, e2, e3, e4, e5, e6, e7, e8, e9 and e10). The inclusion of errors is related to the measurement process [32]. It symbolises the error in the effect of the exogenous variable on the endogenous variable. The significance level was set at two levels: *p* ≤ 0.05 and another at *p* ≤ 0.001. The one-way and two-way arrows are interpreted thanks to the regression weights.

Bentler’s criteria [32] have been followed to develop the model. To assess the fit of the model, quality of fit must be taken into account. The quality of fit is assessed through the chi-square ratio/degrees of freedom (χ^2^/df), comparative quality of fit index (CFI), Tucker–Lewis Index (TLI) and the normalised fit index (NFI). For these indexes, the adjustment value has to be over 0.900 [33]. The root mean square approximation value (RMSEA) index has also been included [32,33,34]. Its value must be less than 0.100 [33,34]. As an overall assessment, the model obtained good values for each of the indices. Table 2 shows the exact values of the indices.

## 3. Results

Table 3 shows the comparative study of the variables according to the sex of the participants (Objective 1 and Hypothesis 1). It is observed that women show higher levels of stress (36.99 ± 7.87) than men (33.06 ± 9.86).

It is observed that men show higher scores in depersonalisation (17.15 ± 6.30) and personal fulfilment (25.02 ± 8.67). On the contrary, those of female gender show a higher recognition of emotional exhaustion (38.28 ± 7.70).

The male population shows higher scores in the variables of control and purpose (2.65 ± 0.81), control under pressure (2.77 ± 0.68) and support networks (2.60 ± 0.73). The female pre-service teachers score higher on persistence/tenacity/self-efficacy (2.61 ± 0.83) and spirituality (2.33 ± 0.85).

Table 4 and Table 5 together with Figure 2 and Figure 3 respond to Objective 2 and research Hypotheses 2–9.

Table 4 and Figure 2 show the results for the male population. A negative effect of resilience (RES) on stress (STR) is observed (β = −0.153; *p* ≤ 0.05). Stress (STR) acts positively on study hours (NSH) (β = 0.374; *p* ≤ 0.001). Resilience (RES) applies a positive effect on study hours (NSH) (β = 0.061). Burnout syndrome (BURN) exerts a beneficial effect on stress (STR) (β = 0.724; *p* ≤ 0.001). On the contrary, burnout syndrome (BURN) does not act as a positive effect on the number of study hours (NSH) (β = −0.479; *p* ≤ 0.001).

Table 5 and Figure 3 show the results obtained for the female population. A negative effect of resilience (RES) on stress (STR) is observed (β = −0.006). Stress (STR) has a positive effect on study hours (NSH) (β = 0.251; *p* ≤ 0.001). Resilience (RES) is positively acting on the number of study hours (NSH) (β = 0.013). Burnout syndrome (BURN) exerts a positive effect on stress (STR) (β = 0.855; *p* ≤ 0.001). Finally, burnout syndrome (BURN) has a negative effect on the number of study hours (NSH) (β = −0.275; *p* ≤ 0.001).

## 4. Discussion

The study found that there are differences in the levels of burnout, stress and resilience according to the declared sex of the participants. There are also differences in the effect of the variables according to the sex of the participants. Contextualization is necessary to try to explain these differences.

According to Objective 1 and Hypothesis 1, it is observed that male trainee teachers show lower stress levels than the female population. Different results were found by García−Martínez et al. [35] stating that the female population shows lower stress levels due to higher emotional competence. The study by Manrique−Millones et al. [36] found a greater effect of stress in the male population than in the female population. Statistically significant differences in favor of women were found in stress symptoms [36]. In contrast, higher scores were found in coping strategies for men.

It is observed that the male sex obtains a higher recognition in self-fulfillment and depersonalization. It is also observed that the female sex obtains a higher score in emotional exhaustion. Puertas−Molero et al. [37] affirm that the continuous subjection to stress generated by the academic environment favors the appearance of disruptive states. This favors the degree of depersonalization of the subjects [37]. Melguizo−Ibáñez et al. [38] affirm that the regular practice of physical exercise helps to eliminate negative feelings when academic goals are not achieved. This is due to the secretion of neurotransmitters such as serotonin and dopamine [39]. The study by Williams and Ayres [40] establishes that women tend to abandon physical sports practice earlier than men. This favors the appearance of negative emotional states [40].

Regarding the variables that make up resilience, it is observed that the male sex obtains better results in the areas of control and purpose, control under pressure and support networks. It is observed that the female sex shows better results in persistence/tenacity/self-efficacy and spirituality. The implementation of intervention programmes has helped the emotional formation of pre-service teachers. Palomera et al. [41] conducted an intervention programme that helped to improve resilience in pre-service teachers. Different resilient behaviors have been observed to address different emotional states. The female sex is more prone to engaging in mental activities such as yoga or tahichí [42]. This is related to a higher degree of spirituality [42]. In contrast, the male gender seeks support from the closest core of peers [42].

Structural equation modeling shows a negative relationship between resilience and stress in both sexes. A greater effect is found for males (Objective 2 and Hypothesis 6). In view of these results, the study by de Vera−García and Gabari−Gambarte [4] affirms that resilience acts are a favorable element for stress channeling. In addition, studies have shown that there are elements that can increase the channeling of disruptive states such as an active and healthy lifestyle [38,43]. It has been observed that the male sex presents a greater duration of physical activity [44]. Through sports practice, peer relationships are fostered [45], and this can lead to a decrease in stress levels [42].

Continuing with the effect of stress on the number of hours of study, a positive effect is realised for both populations (Objective 2 and Hypothesis 3). The effect is greater for the male population (Objective 2 and Hypothesis 7). Very similar results were found by Sánchez−Conde et al. [46], stating that in the preparation for an evaluative test, stress levels increase as the number of hours of study and the pressure that the person exerts on him/herself increase. With respect to the difference found according to sex, a lower effect is observed for the female population. Different results were found by Carmen et al. [47]. In this study, a greater emotional competence was found for the female sex. This leads to a lower presence of stress in the academic life of female pre-service teachers [47].

A negative effect of burnout on the number of study hours is observed for the whole sample (Objective 2 and Hypothesis 4). A greater effect is obtained for the male population (Objective 2 and Hypothesis 8). Very similar results were found by Agyapong et al. [48]. In this study [48], it was found that the male sex shows a higher significance due to lower emotional competence than the female population. Different results were obtained by Melguizo−Ibáñez et al. [2]. This study [2] shows lower levels of burnout syndrome than the female gender due to the practice of physical activity.

A positive effect of resilience on the number of hours of study is observed (Objective 2, Hypothesis 5). The effect is stronger for males (Objective 2, Hypothesis 9). In the field of teaching, resilience plays a key role [19]. Pre-service teachers certainly need an adequate resilient attitude [22]. During the preparation process, the presence of negative emotional states is very prevalent [22]. This leads to a decrease in the quality of the preparation process [22]. With regard to the differences found according to gender, it is observed that previous educational experiences condition the resilient attitude [49]. It has been observed that the emotional competence of the subject conditions how they cope with negative emotional stimuli [50]. In line with these results, Sanabrias−Moreno et al. [51] affirm that the male sex shows greater life satisfaction. This behaviour has a positive effect on the development of resilient behaviour [51].

## 5. Limitations and Future Perspectives

Although the research has met the research objectives and hypotheses, it has a number of limitations, which are shown below.

One of the most important limitations of this study lies in its design, as it is a cross-sectional study, which only allows cause−effect relationships to be established at that point in time. Another limitation is the homogeneity of the sample, as more than two thirds are female. The most outstanding difficulties have been related to reaching a high number of participants. Many responses had to be discarded because they did not meet the inclusion criteria. In terms of strengths, the sample achieved is representative of the Spanish territory.

The future perspectives developed from this study are based on the development of a longitudinal study, in which an intervention programme is used to control the resilience variable, in order to study the effects over a period of time. It is also planned to introduce new variables such as the time of physical activity and the type of physical exercise performed. In addition, it would be desirable to obtain a larger study sample to significantly reduce sampling error. Finally, it would be interesting to include the number of times the entrance exam has been taken.

## 6. Conclusions

Objective 1 and the research hypothesis have been fulfilled. The research concludes that the female population has higher levels of stress. In terms of burnout syndrome, women show a higher level of emotional exhaustion. In contrast, male participants show higher levels of self-realisation and depersonalisation. Still regarding resilience, differences are also observed. The male gender shows higher scores on the variables control and purpose, control under stress and adaptability and support networks.

From the adequate development of the model, it is possible to give an answer to Objective 2 and to the different hypotheses.

In the continuation of Hypothesis 2, this hypothesis has been affirmed. Both structural equation models reflect a negative effect of resilience on stress. Hypothesis 6 has been fulfilled. A greater negative effect of resilience on stress is found for males. Hypothesis 3 was not fulfilled. A positive effect of stress on study hours was obtained for both genders. Hypothesis 7 was fulfilled. A greater effect of stress on study hours is obtained for the male gender. Hypothesis 4 has been fulfilled as stated. A negative effect of burnout on access preparation is observed. On the other hand, Hypothesis 8 was not fulfilled. A greater negative effect is noted for male participants. Hypothesis 5 is fulfilled. A positive effect of resilience on study hours is obtained for both groups. On the other hand, Hypothesis 9 was rejected, as the effect between the above variables is greater for the male participants.

Finally, it is concluded that gender is a key factor in the control of key variables that directly affect the process of competitive examinations for the Spanish civil-service teaching profession. These results also invite further research on psychosocial differences according to the gender of the participants. This study provides a model for understanding psychosocial variables as a function of sex. For future research, it would be interesting to understand these differences. This will make it possible to target intervention programmes in a more comprehensive way.

## Figures and Tables

**Figure 1 behavsci-13-00946-f001:**
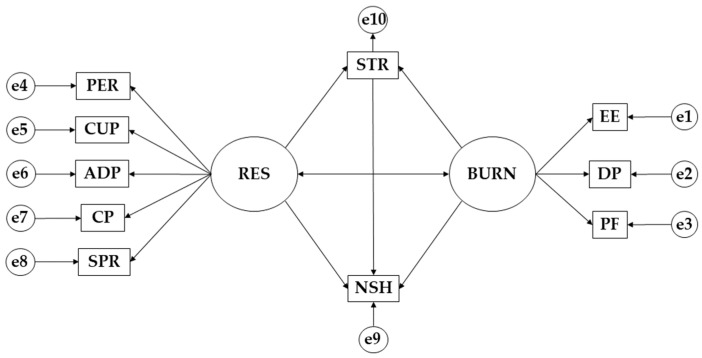
Theorethical model. Note: personal fulfillment (PF); emotional exhaustion (EE); persistence/tenacity/self-efficacy (PER); stress (STR); depersonalization (DP); spirituality (SPR); control and purpose (CP); adaptability and support networks (ADP); resilience (RES); burnout syndrome (BURN); control under pressure (CUP); and number of study hours (NSH).

**Figure 2 behavsci-13-00946-f002:**
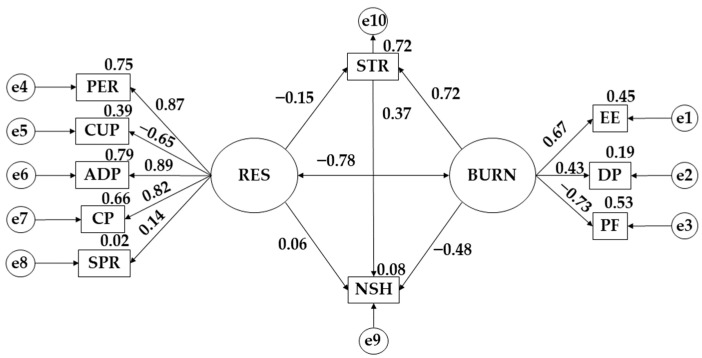
Theoretical model with standardised regression weights for the male gender. Note: emotional exhaustion (EE); persistence/tenacity/self-efficacy (PER); personal fulfillment (PF); stress (STR); depersonalization (DP); adaptability and support networks (ADP); spirituality (SPR); control and purpose (CP); resilience (RES); burnout syndrome (BURN); control under pressure (CUP); and number of study hours (NSH).

**Figure 3 behavsci-13-00946-f003:**
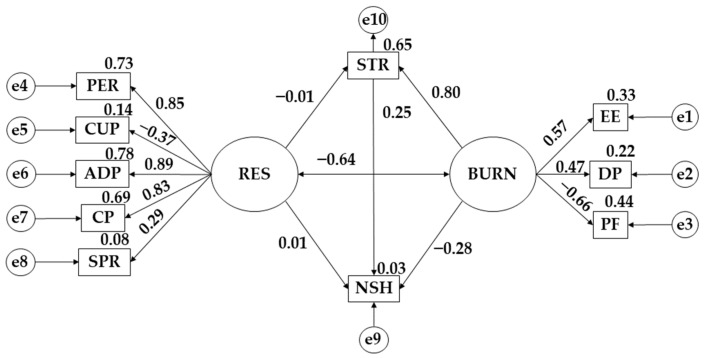
Theoretical model with standardised regression weights for the female gender. Note: personal fulfillment (PF); emotional exhaustion (EE); persistence/tenacity/self−efficacy (PER); stress (STR); depersonalization (DP); spirituality (SPR); control and purpose (CP); adaptability and support networks (ADP); resilience (RES); burnout syndrome (BURN); control under pressure (CUP); and number of study hours (NSH).

**Table 1 behavsci-13-00946-t001:** Characteristics of the study sample.

		N	%
Educational Stage	Early Childhood Education	568	13.8
	Elementary Education	2318	56.3
	High School Education	1231	29.9
Gender	Male	1363	33.1
	Female	2754	66.9

**Table 2 behavsci-13-00946-t002:** Value of the adjustment indices.

χ^2^/df	RMSEA	CFI	TLI	NFI
5.531	0.027	0.900	0.953	0.924

**Table 3 behavsci-13-00946-t003:** Comparative study of the sample regarding gender.

		Levene Test				T-Test			ES (d)	95% CI
		M	SD	F	Sig	T	df	*p*		
STR	Male	33.06	9.86	118.018	0.000	−13.285	2245	≤0.05	0.458	[0.392; 0.524]
	Female	36.99	7.87							
EE	Male	36.01	8.73	29.041	0.000	−8.476	2433	≤0.05	0.282	[0.217; 0.347]
	Female	38.28	7.70							
DP	Male	17.15	6.30	2.973	0.085	4.576	2792	≤0.05	0.143	[0.078; 0.208]
	Female	16.18	6.50							
PR	Male	25.02	8.67	15.311	0.000	4.922	2441	≤0.05	0.163	[0.098; 0.228]
	Female	23.71	7.67							
CP	Male	2.65	0.81	8.941	0.003	7.085	2514	≤0.05	0.249	[0.184; 0.314]
	Female	2.46	0.74							
PER	Male	2.33	0.92	24.032	0.000	−9.785	2499	≤0.05	0.325	[0.260; 0.391]
	Female	2.61	0.83							
CUP	Male	2.77	0.68	5.839	0.016	6.753	2592	≤0.05	0.230	[0.164; 0.295]
	Female	2.62	0.64							
ADP	Male	2.60	0.73	0.286	0.593	7.665	2682	≤0.05	0.263	[0.198; 0.328]
	Female	2.41	0.72							
SP	Male	2.29	0.87	0.339	0.560	−1.315	2655	>0.05	0.513	[0.018; 0.112]
	Female	2.33	0.85							
NHS	Male	5.05	2.50	29.879	0.000	1.968	2491	≤0.05	0.518	[0.001; 0.129]
	Female	4.90	2.27							

**Table 4 behavsci-13-00946-t004:** Standardised regression weights for male gender.

Effect Direction	R.W.				S.R.W.
	Estimates	S.E.	C.R.	*p*	Estimates
**STR** **← RES**	−12.247	4.767	−2.569	**	−0.153
**STR** **← BURN**	1.220	0.104	11.751	***	0.724
**EE** **← BURN**	1.000				0.671
**DP** **← BURN**	0.466	0.032	14.393	***	0.434
**PF** **← BURN**	−1.081	0.047	−22.801	***	−0.731
**PER** **← RES**	1.000				0.140
**CUP** **← RES**	4.890	0.979	4.994	***	0.815
**ADP** **← RES**	4.921	0.983	5.005	***	0.889
**CP** **← RES**	−4.671	0.945	−4.944	***	−0.628
**SPR** **← RES**	5.739	1.147	5.002	***	0.867
**NHS** **← STR**	0.095	0.019	5.029	***	0.374
**NHS** **← RES**	1.232	1.231	1.001	0.317	0.061
**NHS** **← BURN**	−0.205	0.047	−4.404	***	−0.479
**RES** **←→ BURN**	−0.561	0.116	−4.825	***	−0.779

Note: *** *p* ≤ 0.001; ** *p* ≤ 0.05.

**Table 5 behavsci-13-00946-t005:** Standardised regression weights for female gender.

Effect Direction	R.W.				S.R.W.
	Estimates	S.E.	C.R.	*p*	Estimates
**STR** **← RES**	−0.207	1.030	−0.201	0.840	−0.006
**PER** **← RES**	1.434	0.081	17.634	***	0.801
**CUP** **← RES**	1.000				0.287
**ADP** **← RES**	2.469	0.169	14.604	***	0.833
**CP** **← RES**	2.336	0.159	14.695	***	0.886
**SPR** **← RES**	−1.270	0.106	−11.962	***	−0.373
**STR** **← BURN**	2.604	0.178	14.647	***	0.855
**EE** **← BURN**	1.000				0.571
**DP** **← BURN**	0.701	0.036	19.578	***	0.474
**PF** **← BURN**	−1.156	0.047	−24.631	***	−0.66
**NHS** **← RES**	0.122	0.299	0.408	0.683	0.013
**NHS** **← BURN**	−0.142	0.035	−4.030	***	−0.275
**NHS** **← STR**	0.072	0.014	5.093	***	0.251
**RES** **←→ BURN**	−0.692	0.058	−11.845	***	−0.640

Note: *** *p* ≤ 0.001.

## Data Availability

The data used to support the findings of the current study are available from the corresponding author upon request.

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
