# Peer review of "Stress, Resilience, Burnout and Study Hours in Physical Education Pre-Service Teachers—An Explanatory Model about Gender"

_behavsci, 2023, doi:10.3390/bs13110946_

Round 1

Reviewer 1 Report

Comments and Suggestions for Authors

The article presents an appropriate and interesting topic and approach for the journal Behavioral Sciences, addressing contingent issues such as stress, resilience and Burnout Syndrome in teachers in training. However, the authors are requested to consider the following aspects with the idea of improving the manuscript:

-Incorporate in the abstract the proposed structural equation model and the main findings of the study, prior to the conclusions.

-It is suggested that the hypotheses be grouped according to the objectives of the study.

-Specify population and sample, where the students in training come from, inclusion and exclusion criteria, approach and contact with the participants, signature of informed consent when and how and finally who answer the instruments.

-The results and discussion section is very appropriate.

-In limitations and future perspectives, to point out the difficulties and successes in the field work.

-In conclusions, to address the achievement of the hypotheses raised according to the objectives of the research.

-The references are adequate and up to date.

Author Response

REVIEWER 1

Comment 1

Incorporate in the abstract the proposed structural equation model and the main findings of the study, prior to the conclusions

Response 1

Thank you very much for your comment. The results have been modified and the results have been reformulated.

Comment 2

It is suggested that the hypotheses be grouped according to the objectives of the study.

Response 2

Thank you very much for your comment. The research hypotheses have been organized according to the objectives.

Comment 3

Specify population and sample, where the students in training come from, inclusion and exclusion criteria, approach and contact with the participants, signature of informed consent when and how and finally who answer the instruments

Response 3

Thank you very much for your comment. The requested information has been added

Comment 4

The results and discussion section is very appropriate.

Response 4

Thank you very much for your comment.

Comment 5

In limitations and future perspectives, to point out the difficulties and successes in the field work

Response 5

Thank you very much for your comment. The strengths and difficulties have been added.

Comment 6

In conclusions, to address the achievement of the hypotheses raised according to the objectives of the research.

Response 6

Thank you for your suggestion for improvement. A relationship between the research objectives and hypotheses has been made to write the conclusions.

Comment 7

The references are adequate and up to date

Response 7

Thank you very much for your comment.

Reviewer 2 Report

Comments and Suggestions for Authors

First of all, thank you for the opportunity to review this article.

I think it is relevant for knowledge at the level of a professional category, with practical impact.

The article in this form has some limitations, for which I think that some changes can bring added value. therefore, I consider the following observations to be taken into account:

- the introduction of some detailed information in the Abstract (statistical data obtained, enumeration of the instruments used). the number of words limit is not reached.

- identifying/changing keywords. I propose to consider: PE teacher, Spanish educational system

- in the Introduction: 1. The concept of "pre-service teacher" is not clearly defined, 2. The level of knowledge in relation to stress, resilience and well-rout for PE teachers is insufficient, almost non-existent in the presentation. Studies must be identified to highlight if there are differences not only in this professional category, but especially for PE teachers.

- the objectives and hypotheses must be reformulated (the objectives must be SMART: s – specific, m – measurable, a – adaptable, r – achievable, t - temporal), and the hypotheses should either be in the form of assumptions. in the end, it is not clear if all the objectives and hypotheses have been achieved/confirmed.

- the inclusion of some details regarding the number of hours of study (throughout the article, no number is ever referred to)

- line 94 - does "primary education stage" mean "elementary education stage"?

- the source [29] is not found in the text. Likewise, Figure 1. should be referred to in the text where required.

- many bibliographic sources (both from the Introduction and from the Discussions) refer to the level of stress, resilience and burnout (not all the time for teachers, and almost never for PE teachers) during the period of the COVID-19 pandemic (see sources 9,13, 16,17,33,34,42,44,45). Therefore, it should be clearly established if this study covers this period, which is clearly different from other periods. Also, it is not clear whether the study aims to evaluate these processes during the period before the entrance exam to the system or during the school year. It is required, in the procedure, to specify the data collection period, the period when the study was carried out.

- the formula "very distant results" is used several times (see lines 231, 276, 250, 252). What does it mean for authors? Can other forms be identified?

- in Discussions, the authors make comparisons without specifying whether the studies refer to teachers, PE teachers, in particular. Besides, what connection do sources 41 and 42 have with the present study? Therefore, discussions, comparisons and possible explanations should be in accordance with the type of subjects. Sources must be identified that clearly refer to this socio-professional category, with the specifics of the activities.

- in the title, at the end, there is a punctuation mark that must be removed.

- can experience in the profession be an invoked variable that can influence the levels of stress, resilience and burnout in this profession - physical education teacher? it can be mentioned as a limitation of the study if it was not taken into account in the analysis.

Author Response

REVIEWER 2

Comment 1

The introduction of some detailed information in the Abstract (statistical data obtained, enumeration of the instruments used). the number of words limit is not reached.

Response 1

Thank you very much for your suggestion. The suggested changes have been added to the Abstract.

Comment 2

identifying/changing keywords. I propose to consider: PE teacher, Spanish educational system

Response 2

Thank you very much for your suggestion. New keywords have been added

Comment 3

in the Introduction: 1. The concept of "pre-service teacher" is not clearly defined, 2. The level of knowledge in relation to stress, resilience and well-rout for PE teachers is insufficient, almost non-existent in the presentation. Studies must be identified to highlight if there are differences not only in this professional category, but especially for PE teachers.

Response 3

Thank you very much for your suggestion for improvement. The authors have clarified the definition of pre-service teacher. With regard to the relationship between stress, resilience and resilience, the relationship has been extended by extrapolating to the area of pre-service teachers. Several searches of the established concepts have been carried out in the pre-service teacher population. No research articles related to this topic were found in the study population. Nevertheless, the authors have tried to focus the articles with previous research as similar as possible to the study population.

Comment 4

the objectives and hypotheses must be reformulated (the objectives must be SMART: s – specific, m – measurable, a – adaptable, r – achievable, t - temporal), and the hypotheses should either be in the form of assumptions. in the end, it is not clear if all the objectives and hypotheses have been achieved/confirmed.

Response 4

Thank you very much for your comment. The objectives and hypotheses have been reformulated following the information provided.

Comment 5

The inclusion of some details regarding the number of hours of study (throughout the article, no number is ever referred to)

Response 5

Thank you for your interest. In this case this question was measured through an open response where participants had to answer numerically. The response was free and was not categorized. In the structural equation model we work with mean values. Therefore, the mean values of all the variables were used. Table 2 shows the mean value of hours of study according to gender. If a descriptive mean value had been included, it would have been logical to include the mean value of this variable.

Comment 6

line 94 - does "primary education stage" mean "elementary education stage"?

Response 6

Thank you very much for your comment. Primary Education Stage refers to elementary education stage. To avoid possible errors, it has been replaced by elementary education stage

Comment 7

the source [29] is not found in the text. Likewise, Figure 1. should be referred to in the text where required.

Response 7

Thank you very much for your comment. The citation of figure 1 has been added to the text. Reference 29 has also been added.

Comment 8

many bibliographic sources (both from the Introduction and from the Discussions) refer to the level of stress, resilience and burnout (not all the time for teachers, and almost never for PE teachers) during the period of the COVID-19 pandemic (see sources 9,13, 16,17,33,34,42,44,45). Therefore, it should be clearly established if this study covers this period, which is clearly different from other periods. Also, it is not clear whether the study aims to evaluate these processes during the period before the entrance exam to the system or during the school year. It is required, in the procedure, to specify the data collection period, the period when the study was carried out.

Response 8

Thank you very much for your suggestion for improvement. All bibliographic references have been reviewed. As mentioned in previous comments, we have not found any studies on pre-service physical education teachers. However, we have tried to contextualize the study with studies related to this population. Likewise, the study has been carried out during the period prior to the realization of the study.  It has been specified in the text.

Comment 9

the formula "very distant results" is used several times (see lines 231, 276, 250, 252). What does it mean for authors? Can other forms be identified?

Response 9

Thank you very much for your suggestion for improvement. The expression "Very Distant Results" has been removed and replaced

Comment 10

in Discussions, the authors make comparisons without specifying whether the studies refer to teachers, PE teachers, in particular. Besides, what connection do sources 41 and 42 have with the present study? Therefore, discussions, comparisons and possible explanations should be in accordance with the type of subjects. Sources must be identified that clearly refer to this socio-professional category, with the specifics of the activities.

Response 10

Thank you very much for your comment. In this case the references 41 and 42 (from now on 44 and 45) are developed in teachers. Specifically, the 44 specifies in its title "Future Physical Education Teachers" while the 45 is developed in "Teachers"). With respect to the content of the subject matter, this is related to anxiety and the practice of physical activity. This is justified by the gender difference. It has been found that the male sex is more physically active than the female sex. This leads to a decrease in anxiety levels.

Comment 11

In the title, at the end, there is a punctuation mark that must be removed.

- can experience in the profession be an invoked variable that can influence the levels of stress, resilience and burnout in this profession - physical education teacher? it can be mentioned as a limitation of the study if it was not taken into account in the analysis.

Response 11

Thank you very much for your comment. The dot at the end has been removed. In reference to your comment, a study has been developed based on the number of calls submitted to the access process. These results show an increase in burnout and stress syndrome and a decrease in resilience as the number of calls submitted increases. Their suggestion has been added to future perspectives.

Reviewer 3 Report

Comments and Suggestions for Authors

I thank you for the opportunity to review this manuscript. It is an interesting case with a good sample size to develop the study and a pertinent topic to our times.

I am attaching the reviewed manuscript with my comments. Among the most important ones are:

1. Please, conduct an English Language Professional Editing process.

2. Present the research questions derived from the objectives and followed by the corresponding hypotheses.

3. Organize the literature review more comprehensively to support your hypotheses. The reviewed literature must keep the actual relationships shown in the model. It is recommendable to present this evidence separately for those relevant relationships between constructs.

4. Justify adequately, why the statistical procedures (Levene and t-test) were chosen to complete the structural model and present these inferences properly.

5. Present all the goodness of fit and reliability indexes in a clearer manner and the reference cut-off values as well for each one according to the AMOS and literature reviewed.

6. It is recommended to conduct a more complete discussion of the results comparing these with those findings from other authors and professions.

Comments on the Quality of English Language

1. It is recommended to submit to a professional English Editing to improve the readability and comprehension of the manuscript.

Author Response

REVIEWER 3

Comment 1

  1. Please, conduct an English Language Professional Editing process.

Response 1

Thank you very much for your comment. The wording of the complete text has been reworded.

Comment 2

  1. Present the research questions derived from the objectives and followed by the corresponding hypotheses.

Response 2

Thank you very much for your proposal for improvement. A reviewer previously told us to relate the objectives to the hypotheses. To satisfy his suggestion, the authors have placed the different research questions followed by the objectives and hypotheses of the study.

Comment 3

  1. Organize the literature review more comprehensively to support your hypotheses. The reviewed literature must keep the actual relationships shown in the model. It is recommendable to present this evidence separately for those relevant relationships between constructs.

Response 3

Thank you very much for your suggestion. The authors have structured the theoretical framework as follows:

Paragraph 1: Explanation of the teacher selection process.

Paragraph 2: Effects of the selection process at the mental level. Contextualization of stress.

Paragraph 3: Theoretical contextualization of the "Burnout Syndrome".

Paragraph 4: Theoretical contextualization of "Resilience".

Paragraph 5: Current status of the variables in the population under study.

The authors consider that the theoretical review used for each of the paragraphs has been adequate. In view of your comments, we have proceeded to carry out a more current and more appropriate review.

Comment 4

  1. Justify adequately, why the statistical procedures (Levene and t-test) were chosen to complete the structural model and present these inferences properly.

Response 4

Thank you very much for your comment. The authors have more fully developed the choice of these statistical tests.

Comment 5

  1. Present all the goodness of fit and reliability indexes in a clearer manner and the reference cut-off values as well for each one according to the AMOS and literature reviewed.

Response 5

Thank you very much for your comment. The cut-off values have been specified. Thank you very much for your comment. The cut-off values have been specified. Likewise, the cut-off values have been previously referenced by expert authors in structural equations.

Comment 6

  1. It is recommended to conduct a more complete discussion of the results comparing these with those findings from other authors and professions.

Response 6

Thank you very much for your comment. The discussion of the results has been modified. Some citations have been replaced because they did not match the study population.

Comment 7

Please, explain separately those relationships relevant to your theory taking into account the main variables of your study.

How resilience and burnout have been related by other authors and all other variables of interest of your study?: stress, gender, number of hours of study and burnout

Response 7

Thank you very much for your suggestion. The authors have structured the theoretical framework as follows:

Paragraph 1: Explanation of the teacher selection process.

Paragraph 2: Effects of the selection process at the mental level. Contextualization of stress. Relationship between stress and burnout syndrome.

Paragraph 3: Theoretical contextualization of the "Burnout Syndrome". Relationship between burnout syndrome and resilience

Paragraph 4: Theoretical contextualization of "Resilience".

Paragraph 5: Current status of the variables in the population under study.

The authors consider that the theoretical review used for each of the paragraphs has been adequate. In view of your comments, we have proceeded to carry out a more current and more appropriate review.

Comment 8

Please, present in a more organizade manner all the goodnes of fit indexes and reliability and other indices for each construct of your model and their reference values.

Response 8

Thank you very much for your suggestion. The information has been organized in a clearer way.

Comment 9

It is recommendable to include a general objective of the manuscript. In addition, it is more relevant to present the research questions for which the present hypotheses corrrespond.

Response 9

Thank you very much for your comment. The authors consider that this research has three general objectives. Derived from each of them are different research hypotheses. The study is developed through these three general objectives. The authors consider that to agglutinate these three objectives into one would make it difficult to understand the text. Your suggestion will be taken into account for future studies.

Comment 10

Please, explain how these instruments were selected as instruments for gathering data?

Response 10

Thank you very much for your comment. The requested section has been developed.

Comment 11

Present the theoretical evidence of the established relationship or your structural models in a more organized manner to improve the comprehension of your models.

Response 11

Thank you very much for your suggestion. The introductory section has been reworded.

Reviewer 4 Report

Comments and Suggestions for Authors

Congratulations for your work (great sample and analysis). I hope these observations could help you to improve the clarity and congruence of your study. Please, find my comments in the document attached.

Comments on the Quality of English Language

The general quality of english is good, although I would recommend a deeper review since I found some mistakes.

Author Response

REVIEWER 4

Comment 1

and gender?: is not a main variable of this study? When I read the abstract I understand that gender is the core of the research, but this is not addressed in the title

Response 1

Thank you very much for your comment. The word gender has been added to the title.

Comment 2

Rewrite this sentence clearer

Response 2

Thank you very much for your comment. The wording of the requested sentence has been reworded.

Comment 3

I think the grammar/redaction could be easier/simpler.

Response 3

Thank you very much for your comment. The discussion of the results has been modified. The wording of the research has been reformulated.

Comment 4

what is the need for this study? what novelty this reserach brings?

Response 4

Thank you very much for your comment. The novelty of this article has been specified.

Comment 5

is this objective answered? effect of stress on study hours?

Response 5

Thank you very much for your comment. The wording of objective 2 has been reworded.

Comment 6

The hypothesis should indicate the direction of the differences

Response 6

Thank you very much for your comment. The research hypotheses have been redrafted. The direction in which the effect occurs has been specified.

Comment 7

"males tend to have an earlier onset of this disruptive state, due to lower emotional formation (Line 57)". Does the introduction say the opposite to your hypothesis?

Response 7

Thank you very much for your review. After looking at hypothesis 5, it has been noted that it was poorly formulated. This hypothesis relates to the statement (males tend to have an earlier onset of this disruptive state, due to lower emotional formation).

H.5.The effect of burnout syndrome will be positive on stress for the male sample.

Due to the low emotional competence presented by the male sex, it is expected that stress will act positively on the effect of burnout syndrome.

Comment 8

Too repetitive

Response 8

Thank you very much for your comment. The wording of the article has been revised. Redundancy has been eliminated.

Comment 9

Detail each abbreviations

Response 9

Thank you very much for your comment. Abbreviations have been removed. The name of each variable has been set out.

Comment 10

"who". I recommend to review english grammar and redaction

Response 10

Thank you very much for your comment. The wording of the article has been revised

Comment 11

Include the explanation of: e1, e2, e3...

Response 11

Thank you very much for your comment. In the data analysis section it has been specified what the symbols refer to.

Comment 12

could you discussed each of the dimensions analysed?

Response 12

Thank you very much for your comment. The authors have conducted a new literature review. We have proceeded to search for research that discussed each of the dimensions. Despite our efforts, we have managed to add some research citing some of the dimensions.

Comment 13

is the relation between these 2 variables embrace in the hyphotesis?

Response 13

Thank you very much for your comment. The research hypotheses have been reformulated. The authors believe that they are now better worded. In addition, the hypotheses have been grouped according to the study objective

Comment 14

The argument of the importance of being active is too repetitive in the discussion. Have you thought in other possible reasons?

Response 14

Thank you very much for your comment. We have been looking for new reasons to stop being repetitive. Despite this, we have maintained the importance of being physically active.

Comment 15

is this one of the hypothesis of the study?

Response 15

Thank you very much for your comment. All research objectives and hypotheses have been answered.

Comment 16

and what about the gender differences?

Response 16

Thank you very much for your comment. The discussion has been rephrased. New discussion sections related to gender have been added.

Comment 17

Males have lower emotional competence but better resilience? is this congruent?

Response 17

Thank you very much for your comment. The error has been removed

Comment 18

Have each of the research hypothesis been confirmed?

Response 18

Thank you very much for your comment. After reviewing all the research again, all the research objectives and hypotheses have been answered.

Comment 19

Please, review the paper and better align the sections: Objectives & hyphotesis, with results, discussion and conclusions

Response 19

Thank you very much for your comment. A closer link between the requested sections has been made.

Comment 20

are all of the hypothesis addressed on the conclusions?

Response 20

Thank you very much for your comment. all hypotheses have been answered.

Comment 21

review writing

Response 21

Thank you very much for your comment. The wording of the article has been revised

Comment 22

was this a hypothesis of the study?

Response 22

Thank you very much for your comment. all hypotheses have been answered

Comment 23

this affirmation is contrary to the hyphotesis 3

Response 23

Thank you very much for your comment. All research hypotheses have been answered.

Comment 24

could you provide de relevance/the contribution to the field of this study?

Response 24

Thank you very much for your comment. The requested information has been provided

Round 2

Reviewer 2 Report

Comments and Suggestions for Authors

great job!

congrats!

Author Response

Thank you very much for your review